# Clinical Experience with Anti-IgE Monoclonal Antibody (Omalizumab) in Pediatric Severe Allergic Asthma—A Romanian Perspective

**DOI:** 10.3390/children8121141

**Published:** 2021-12-06

**Authors:** Elena Camelia Berghea, Mihaela Balgradean, Carmen Pavelescu, Catalin Gabriel Cirstoveanu, Claudia Lucia Toma, Marcela Daniela Ionescu, Roxana Silvia Bumbacea

**Affiliations:** 1Department of Pediatrics, Carol Davila University of Medicine and Pharmacy, 020021 Bucharest, Romania; camelia.berghea@umfcd.ro (E.C.B.); mihaela.balgradean@umfcd.ro (M.B.); catalin.cirstoveanu@umfcd.ro (C.G.C.); claudia.toma@umfcd.ro (C.L.T.); roxana.bumbacea@umfcd.ro (R.S.B.); 2Department of Pediatrics, “Marie S. Curie” Emergency Children’s Clinical Hospital, 041451 Bucharest, Romania; carmen.pavelescu@rez.umfcd.ro; 3Pneumology Department, “Marius Nasta” Institute of Pneumology, 050159 Bucharest, Romania; 4Allergology Department, “Dr Carol Davila” Nephrology Clinical Hospital, 010731 Bucharest, Romania

**Keywords:** allergic asthma, anti-IgE, Omalizumab, observational study, children

## Abstract

Background: Asthma is the most common chronic disease affecting children, with a negative impact on their quality of life. Asthma is often associated with comorbid allergic diseases, and its severity may be modulated by immunoglobulin E (IgE)-mediated allergen sensitization. Omalizumab is a humanized monoclonal anti-IgE antibody, the first biological therapy approved to treat patients aged ≥6 years with severe allergic asthma. The primary objective of our study was to investigate the efficacy and safety of Omalizumab in Romanian children with severe allergic asthma. Methods: In this observational real-life study, 12 children and adolescents aged 6 to 18 years (mean 12.4 years) with severe allergic asthma received Omalizumab as an add-on treatment. Asthma control, exacerbations, lung function, and adverse events were evaluated at baseline and after the first year of treatment. Results: We observed general improvement in total asthma symptom scores and reduction in the rate of exacerbation of severe asthma. Omalizumab treatment was associated with improvement in the measures of lung function, and no serious adverse reactions were reported. FEV1 improved significantly after one year of treatment with Omalizumab [ΔFEV1 (% pred.) = 18.3], and [similarly, ΔMEF50 (%) = 25.8]. The mean severe exacerbation rate of asthma decreased from 4.1 ± 2.8 to 1.15 ± 0.78 (*p* < 0.0001) during the year of treatment with Omalizumab. Conclusions: This study showed that Omalizumab can be an effective and safe therapeutic option for Romanian children and adolescents with severe allergic asthma, providing clinically relevant information on asthma control and exacerbation rate in children and adolescents. The results demonstrated the positive effect of Omalizumab in young patients with asthma, starting from the first year of treatment.

## 1. Introduction

Asthma is defined as a chronic inflammatory disease of the lower respiratory tract; it affects between 1 and 18% of the general population, usually starting in early childhood [1,2,3]. On average, of the global population, between 10 and 12% of children under the age of 6 or 7 years, and 14% of adolescents between the ages of 7 and 14 years suffer from asthma [4,5]. Up to 90% of childhood asthma has an allergic background and is associated with a personal or family history of other allergic diseases such as allergic rhinitis, atopic dermatitis, or food allergy. Children with other allergies have a 30% increased risk of developing asthma [6,7].

According to the Global Initiative for Asthma (GINA) five-step guidelines, asthma is considered severe (GINA step 4–5) if high doses of controller therapy are needed for control, if it remains uncontrolled despite correct administration of therapy and treatment of contributory factors, or if the asthma worsens with the step-down of doses of treatment [1]. Following systematic assessment to optimize asthma control [6,7,8], in approximately 5–15% of patients, the asthma continues to be severe [9], a condition that is associated with increased morbidity and mortality. The incidence of severe pediatric asthma is about 2.5% of all children with asthma [8,9]. It accounts for approximately half of the total health resources for pediatric asthma, and it is associated with a higher likelihood of exacerbation and an increased risk of mortality and of chronic obstructive pulmonary disease (COPD) in adulthood [8,9]. Safe and effective treatment options are, therefore, considered necessary for children with uncontrolled severe asthma [10].

The discovery of immunoglobulin E (IgE) as a key player in allergic diseases, and the association of higher mean plasma levels of IgE with severe asthma [11], provided the rationale for developing safe and effective targeted anti-IgE therapy. Anti-IgE therapeutic intervention has now become an important option for both adults and children with severe uncontrolled allergic asthma [12,13,14]. Asthma is a mixed syndrome combining several immunological subtypes [15,16,17], of which the most comprehensively described is the allergic phenotype. The development of allergic asthma involves mainly the synthesis of IgE antibodies against aeroallergens, such as mold, cat and dog dander, mites, etc. The ability to generate high levels of specific IgE antibodies is favored by the genetic susceptibility for such a response and is accompanied by imbalance in the ratio of T-helper 1 (Th-1) to T-helper 2 (Th-2) lymphocytes (Th1/Th2). One characteristic of allergic inflammation, the higher expression of Th-2 cytokines than Th-1 cytokines, results in the production of various interleukins (ILs), specifically IL-4, IL-5, IL-6, IL-9, and IL-13, which act as allergic inflammation mediators [18,19]. The high incidence of allergy in pediatric asthma, and the increased levels of serum total IgE usually encountered in severe forms of the disease, are sound reasons for the use of anti-IgE therapy in children. Omalizumab, a subcutaneously administered humanized anti-IgE monoclonal antibody, has been approved as add-on therapy for patients with moderate to severe persistent allergic asthma that remains uncontrolled despite daily high-dose inhaled corticosteroids (ICS) plus inhaled long-acting beta-adrenoceptor agonist (LABA) treatment or other controller treatment, who have a positive skin prick test response or in vitro reactivity to a perennial aeroallergen [20,21,22]. Owing to the central role of IgE antibody in the pathophysiology of allergic disorders, Omalizumab was the first biological agent to be authorized for the treatment of severe allergic asthma in children. Omalizumab is a murine (95%) humanized recombinant IgG1 monoclonal antibody. Its mechanism of action consists in the selective binding of free IgE and inhibition of the interaction of IgE with the high-affinity IgE receptor (FcεRI)-localized mast cells and basophils, thus disrupting the allergic cascade and influencing the entire inflammatory process, through reduction of activation of inflammatory cells and decrease in the release of pro-inflammatory factors (early phase of allergic response) [5,22]. In addition, by attaching to FcεRII/CD23 receptors on B cells, Omalizumab affects their antigen-presenting role, which modulates the dialogue between B cells and T cells and induces significant (≤99%) downregulation of IgE receptors and rapid reduction of free IgE levels, thereby blocking Th2 amplification of the inflammatory response. Prevention by Omalizumab of degranulation of the mast cells and basophils reduces the inflammatory response in the upper and lower airways in all phases of the allergic reaction, also enabling modulation of the process of airway remodeling [6], thereby effectively improving the control of symptoms and retarding the progression of asthma [23,24].

As outlined in a wide range of clinical trials and real-life studies, Omalizumab has been widely demonstrated to be a clinically effective and safe treatment for adults and children, reducing asthma exacerbation rates, allowing reduction in the doses of corticosteroids needed to control asthma symptoms, and improving pulmonary function tests (PFTs) and the quality-of-life scores [20].

Our study aimed to evaluate, for the first time to our knowledge, the effect of Omalizumab in severe pediatric allergic asthma, to investigate whether the outcome of treatment in a Romanian population is consistent with the findings published from other countries.

## 2. Materials and Methods

We performed an observational retrospective study on hospitalized inpatient pediatric patients who met the criteria for severe allergic asthma for which they had and received add-on therapy with Omalizumab for at least 12 months. This single center study was conducted in the M.S. Curie Children’s Emergency Clinical Hospital in Bucharest, Romania, between January 2013 and December 2020. The following inclusion criteria were used: a documented diagnosis of asthma with criteria for severe allergic asthma (see below), age at the start of Omalizumab treatment between 6 and 18 years (to comply with Romanian regulations), and at least 12 months of continuous treatment with Omalizumab at the start of the study. The subjects with less than 12 months of continuous Omalizumab therapy and those with missing or incomplete data at 12 months of treatment were excluded.

Criteria for severe asthma in children and adolescents have been defined according to the current GINA guidelines [1] as asthma requiring stage 4 or 5 of treatment to get control of symptoms or remaining uncontrolled despite the high level of optimized treatment. In order to ensure the comparability between all the study subjects, we selected the first year of treatment with Omalizumab for those patients who had undergone their 1st year of treatment with Omalizumab in subjects with longer than one year treatment.

In Romania, Omalizumab is indicated for children (aged 6 to <12 years) and adolescents (aged >12–18 years) with severe persistent allergic asthma who have a positive skin test or in vitro reactivity (blood test) to a perennial aeroallergen and who have frequent daytime symptoms or night-time awakenings, and who have had multiple documented severe asthma exacerbations despite daily high-dose ICS plus LABA [23]. In addition, in patients aged >12 years, only those with forced expiratory volume in 1s (FEV1) less than 80% predicted were included [23].

All the patients registered in the study hospital who started Omalizumab therapy after 2013 (the year it became available in Romania) were reviewed for the presence of inclusion criteria and the absence of the exclusion criteria. Data were retrieved from the patients’ hospital records: demographic data, age at asthma diagnosis, ICS dosage and medications used, visits to the emergency department, hospitalizations/or intensive care unit admissions for asthma ever, asthma severity, exacerbations requiring the use of systemic corticosteroids, total IgE levels, absolute eosinophil count in peripheral blood (evaluated by standard automated laboratory measurement), and allergic sensitization, estimated by skin prick test and/or specific IgE to common perennial and seasonal allergens (Dermatophagoides pteronyssinus, Dermatophagoides farinae, trees, grasses and weeds pollens, molds, cat and dog dander), and comorbidity data (allergic rhinitis, atopic dermatitis). 

PFTs were performed in all patients, in accordance with the European Respiratory Society/American Thoracic Society (ERS/ATS) guideline criteria [25]. The same computer-based spirometer (Jaeger, Viasys Healthcare GmbH, Hoechberg, Germany) was used at all times, and a volume calibration was performed daily in the morning. All lung function measures were performed in the morning by the same certified technician, and the patients took their controller medications as usual. The criteria of acceptable repeatability were achieved for all the assessments (the difference between the largest and the next largest FVC and FEV1 is ≤0.150 L [25]. Through appropriate coaching, all the patients performed acceptable spirometry, and the best 3 measurements were evaluated (and usually no more than eight maneuvers were necessary). Forced expiratory volume in 1 s (FEV1) and forced expiratory flow at 25–75% of the forced vital capacity (FVC) (FEF25–75%) were expressed as % predicted values, and the FEV1/FVC ratio as a percentage.

Exploration of efficacy included changes in: (1) asthma control level; (2) severe exacerbation rate and healthcare use; (3) lung function over the first year of treatment, i.e., morning FEV1 and forced expiratory flow rate at 50% of forced vital capacity (MEF50); (4) ICS dosage as maintenance therapy. Safety assessment consisted of the listing of adverse events recorded during one year of doses administered.

The responsiveness to Omalizumab was evaluated using as criteria the achievement of asthma control over the year of treatment, reduction in severe exacerbation rate and healthcare use in comparison with that observed before Omalizumab therapy, reduction in the dose needed to maintain treatment, and improvement in PFTs over the year of treatment. 

Statistical analysis was performed using SPSS IBM 26, and statistical significance was set at *p* < 0.05.

## 3. Results

A group of 12 patients aged between 6 and 18 years (mean 12.4 ± 4.1 years) was identified with severe allergic asthma, who fulfilled the inclusion criteria for additional treatment with Omalizumab, which was given between 2013 and 2020. The group was included in a retrospective analysis. All the relevant analysis data were evaluated at initiation of Omalizumab and after the first year of treatment. 

The baseline characteristics of the patients are detailed in Table 1. The study group consisted of three girls (25%) and nine boys (75%), and their mean age at asthma diagnosis was 6.5 ± 3.77 years. The prevalence of severe allergic asthma in pediatric patients registered in database of the M.S. Curie Children’s Emergency Clinical Hospital is 1.79%. History of atopic dermatitis was recorded in 9 patients (75%) and allergic rhinitis in 10 (83.3%). A family history of atopy was reported in eight patients (66.6%). All 12 patients had allergic sensitization to at least one perennial allergen relevant to asthma symptoms, and 11 (91.7%) had allergic sensitization to multiple inhaled allergens.

The mean dose of ICS (fluticasone propionate equivalent) taken by the patients was 468.7 μg/day, and most patients received additional treatment with leukotriene receptor antagonists (LTRA) (91.7%) and LABA (66.7%). In accordance with GINA guideline, we considered as exacerbations the episodes of increasing of asthma symptoms and decrease in lung function requiring a change in treatment [1]. We considered clinically significant an exacerbation that required at least 3 days of systemic corticosteroids. According to these criteria, the study patients had experienced a mean of 4.1 ± 2.8 exacerbations of asthma and, at inclusion, most experienced limitations in daily activities. The level of asthma control at the time of treatment decision with Omalizumab was classified according to the GINA guideline criteria, i.e., controlled, partially controlled, and uncontrolled. According to the recommendations for assessment of symptom control, asthma is well controlled if there are daytime symptoms for not more than two times per week; there is no limitation of activities; there are no nocturnal symptoms/awakenings; and there is need for relief/rescue treatment twice or less per week. Asthma is partially controlled if 1–2 of these criteria are present and not controlled if 3–4 of these criteria are met [1]. Thus, before the initiation of add-on therapy with Omalizumab, among the study patients, the asthma was classified as partially controlled in five patients (41.7%), while in seven patients (58.3%), the disease was not controlled despite high doses of controller treatment.

In order to compare the results of treatment with Omalizumab in the study group and to monitor asthma control, we used the consensus-based GINA symptom control tool alongside with assessment of symptom control by asthma control test, adapted for the age of the patient, specifically, the Asthma Control Test (ACT–ACT) for adolescents, and the Childhood Asthma Control Test (c-ACT) for children 6–11 years of age. The asthma control test is a numerical asthma control assessment that is swell correlated with the GINA asthma control classification levels (GINA). Thus, the control of asthma symptoms was demonstrated to be clearly improved in the first year of treatment with Omalizumab. The asthma control levels after one year were rated as controlled in 75% of cases, partially controlled in 25% of cases, and uncontrolled in none.

A decrease in the mean rate of severe exacerbations was observed: prior to treatment with Omalizumab, all of the patients had recorded more than two asthma exacerbations in the previous year, while, during one year of treatment, only 16.7% of patients had more than two asthma exacerbations induced by exercise, and none had viral-induced exacerbations. The mean rate of severe exacerbations decreased from 4.1 ± 2.8 per patient in the previous year to 1.15 ± 0.78 in the course of the treatment year (*p* < 0.0001).

PFT prior to Omalizumab treatment showed a decrease in FEV1 (mean 86.74%) and MEF50 (mean = 76.30%), which improved after one year of treatment with Omalizumab (mean FEV1 = 105.03%, mean MEF50 = 102.13%), as shown in Figure 1. 

A mean reduction in eosinophil count of 280 ± 166 cells/mL was observed in the patients after one year of treatment with Omalizumab, as shown in Figure 2.

A reduction in the maintenance dose of the medication was possible in nine patients (75%), and ICS use was reduced in all patients. The mean ICS dose at 12 months was lower by the equivalent of 275 μg fluticasone propionate (Table 2).

Recorded safety data for the year of Omalizumab treatment were collected. All the observed reactions were mild, and all appeared after the first dose of medication, with the most frequent being pain at the injection site (12 patients), flu-like symptoms (5 patients), and headache (3 patients). No adverse reaction resulted in discontinuation of therapy, and there were no anaphylaxis reports in the study group during either the first year reported in this study nor for successive years of treatment with Omalizumab.

## 4. Discussion

We have reported here a retrospective observational survey of 12 Romanian children and adolescents with severe allergic asthma, which demonstrated that they benefited from additional anti-IgE monoclonal antibodies (Omalizumab) therapy for high-level maintenance treatment. 

In pediatric asthma, control of symptoms can be achieved in most cases by low to medium doses of ICS, plus one or more controlling drugs. In poorly controlled cases, the cause is often related to technical mistreatment or errors in the administration of the control drugs. Although less frequently than in adults, in some children with severe allergic asthma, the disease remains uncontrolled, despite high doses of standard medications. In clinical studies, complementary treatment with IgE monoclonal antibodies was clinically effective, with a favorable safety profile, in patients with severe asthma [9,26], chronic urticaria [27,28,29], and, more recently, in patients with chronic rhinosinusitis with nasal polyposis [30]. Absorption of Omalizumab into the systemic circulation is relatively slow, with peak serum concentrations achieved after an average of 7–8 days. This monoclonal anti-IgE antibody demonstrates linear pharmacokinetics in approved dosing regimens. The process of clearance of IgG, and the specific and complex binding formation with IgE, are involved in the clearance of omalizumab. The average half-life is 26 days [8].

Reduction of the serum IgE level is observed within a few hours of administration, and the number of high-affinity IgE receptors diminishes following 8–12 weeks of treatment, with peak levels apparent 7–8 days after single-dose administration and steady-state levels reached in the serum about 14–28 days following multiple-dose administration [26].

Observational studies provide real-life data and can add valuable information other than that obtained in clinical trials. To our knowledge, this is the first observational study investigating the efficacy of Omalizumab as add-on therapy to high-level maintenance treatment for children and adolescents in Romania with severe persistent allergic asthma. Over one year of treatment, the rate of clinically significant asthma exacerbations is a good assessment of efficacy of treatment in clinical studies. Three pivotal Phase III clinical trials in patients with moderate to severe asthma were performed in the USA and Europe on a population of 1071 ICS-dependent symptomatic adolescents and adults, and 334 children (aged 6–12 years), followed by 25 trials enrolling a total of 6382 patients with uncontrolled allergic asthma, were analyzed in a Cochrane meta-analysis, which demonstrated effective reduction of asthma exacerbations by Omalizumab treatment [31,32]. Recent systematic reviews of the results of multiple randomized, blinded, placebo-controlled phase III studies involving children aged 6 to 11 years [33] or adults and children aged over 6 years [33,34] also confirmed effective improvement in asthma control by Omalizumab, with reduction in asthma exacerbations, hospital admissions, acute asthma attacks and the related need for oral corticosteroid (OCS) in children with severe asthma. In our study, a statistically significant reduction in the rate of exacerbations during the first year of treatment was observed; the mean rate of severe exacerbations decreased from 4.1 ± 2.8 per patient during the previous year to 1.15 ± 0.78 during the first year of treatment (*p* < 0.0001), and there was a decrease in the number of patients who developed asthma exacerbation, with less acute asthma symptoms induced by exercise (2 out 12 patients) and no exacerbations induced by viral respiratory infections during the first year of treatment with Omalizumab. This result is in concordance with those of other observational studies; for example, Deschildre and colleagues reported a 72% reduction in the mean number of exacerbations over one year and two years of treatment (1.25 vs. 4.4, *p* < 0.0001), in an observational study involving 104 patients with severe allergic, partially/poorly controlled asthma [35,36]; Licari and colleagues documented a 91% reduction of asthma exacerbation in a cohort of 47 patients aged 6–21 years, with severe allergic asthma on Omalizumab: (6–21 years old) by Licari et al. [20]; Pitrez and colleagues showed that hospitalization decreased by 70% (*p* = 0.02) after 6 months of treatment with Omalizumab in a cohort of 14 children and adolescents (6–18 years) with severe allergic asthma [37]; and finally, in a group of 38 Japanese children (6–15 years) with severe persistent allergic asthma, a 69.2% reduction of frequency of exacerbations was recorded after they received Omalizumab [38].

In our study group, the prevalence of asthma was higher in males, which is a finding similar to that of the results from other studies on children and adolescents with severe allergic asthma who have been administered Omalizumab treatment [20,34,35,38] and concordant with the general tendency for a higher prevalence of asthma in boys in childhood [39]. Results from cluster analysis targeting children with severe asthma highlights the heterogeneity of asthma in the pediatric population. A recently published Korean study has reported a predominance of early-onset atopic asthma in males, suggesting that sex hormones are involved differently in the mechanism of asthma, depending on age [40].

It is also important to note that the degree of decrease in the exacerbation rate has been correlated in clinical trials with the number of exacerbations before starting the treatment with Omalizumab, baseline PFTs, and eosinophil count, with different results in different studies, but with improvements being greater in children with more severe subtypes [38]. A raised absolute eosinophil count in peripheral blood is associated with a higher risk of asthma exacerbations [1,33]. In our study, 91.7% of the patients were polysensitized, a trait that, together with the high eosinophil counts and the increased and high levels of total IgE, are parameters related to the severity of asthma. It has been suggested that these characteristics may be related to a subpopulation of children with serious, highly allergic asthma, who will show a good response to Omalizumab [35]. An elevated production of total IgE has been demonstrated to be a marker of asthma severity in children [35].

In our study group, we observed a slight decrease in the mean absolute number of peripheral eosinophil cells after one year of treatment. The effect appears to be correlated with the duration of treatment, as a lower value compared to baseline has recently been reported after 6 years of treatment [34]. The pulmonary function evaluated by both large (FEV1) and small (MEF50) airway functional parameters, improved in our study in children after one year of treatment with Omalizumab. As has been observed in other studies, the mean baseline FEV1 was >80% in our patients, due to the young age of the patients whose lung function has not already reduced, as we would expect in severe asthma in adults [14]. Although the improvement in PFTs may not be clinically relevant, it is discussed in the literature as a useful parameter for assessing the therapeutic effect of a specific treatment, since severe forms of asthma in children have been reported to decrease pulmonary function at long-term follow-up [41,42].

The level of asthma control, another important measure of treatment effectiveness in asthma, improved in our group (75% controlled, 25% partially controlled and non-controlled after the first year of treatment, versus 41.7% partially controlled, 58.3% non-controlled before additional therapy with Omalizumab), similar to data from other observational studies [14,38]. A dose reduction of controlled medication with the maintenance of control of asthma symptoms was possible in nine patients (75%) in our group after the first year of anti-IgE treatment. 

Concerning the safety of Omalizumab, all the patients in our group had at least one adverse event during the treatment, but all of mild severity. The most common adverse reactions reported were transient localized reactions and local pain. The good safety profile of treatment with Omalizumab in children and adolescents with asthma was evident in clinical and observational studies addressing concerns about the potential association of Omalizumab with hypersensitivity to adverse reactions. In this context, it should be noted that no cases of anaphylaxis were reported in children compared with the risk of anaphylaxis of 0.1 to 0.2% reported in adults and adolescents [43,44].

The major limitation of this study is the small number of patients. Our goal is to enroll new patients and continue follow-up of patients recruited beyond one year after administration of Omalizumab treatment, in order to confirm the positive findings of the study. 

The main strength of this article is that it presents the first Romanian observational study reporting the outcome of children and adolescents with severe allergic asthma who received Omalizumab. The results confirm that Omalizumab improved asthma control reduced the rate of exacerbations in this population, and was well tolerated over one year of treatment.

## 5. Conclusions

Understanding the benefits of Omalizumab treatment and the importance of adherence to treatment in the pediatric population may provide insights that may guide the management of Omalizumab in patients with asthma treated with this therapy.

In the Romanian pediatric population, one year of Omalizumab therapy improved control of symptoms of severe asthma and lung function, was correlated with a marked reduction in asthma exacerbations caused by viral infections or by exercise, and permitted reduction in the dosage of controller medication. Based on studies that have reported results in the 24–36 month time frame or at 36 months and beyond, there is compelling evidence to infer the long-term efficacy of Omalizumab in the management of severe allergic asthma. 

In conclusion, Omalizumab represents a pivotal therapeutic option for severe allergic asthma in children and adolescents whose asthma remains uncontrolled despite high doses of ICS and other control drugs. More long-term follow-up studies are needed to confirm these promising results. Data collected through observational real-life studies complement the findings of trials and highlight the efficacy and safety profile of the treatment of severe asthma in children and adolescents with Omalizumab.

## Figures and Tables

**Figure 1 children-08-01141-f001:**
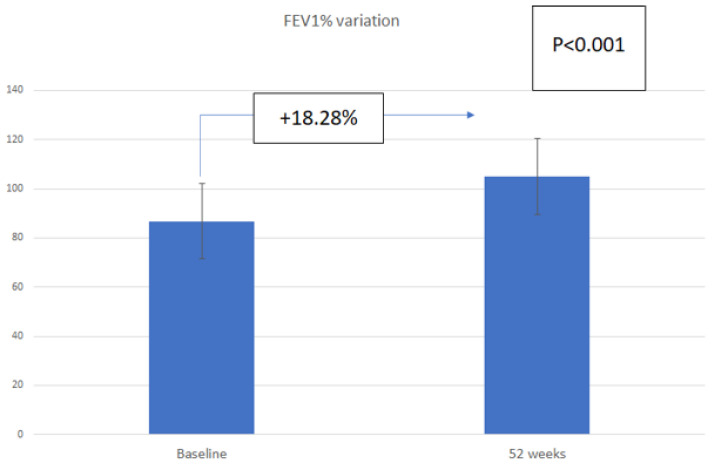
Change in forced expiratory volume in 1 s (FEV1%) in children and adolescents with asthma (n = 12) after 52 weeks of treatment with Omalizumab.

**Figure 2 children-08-01141-f002:**
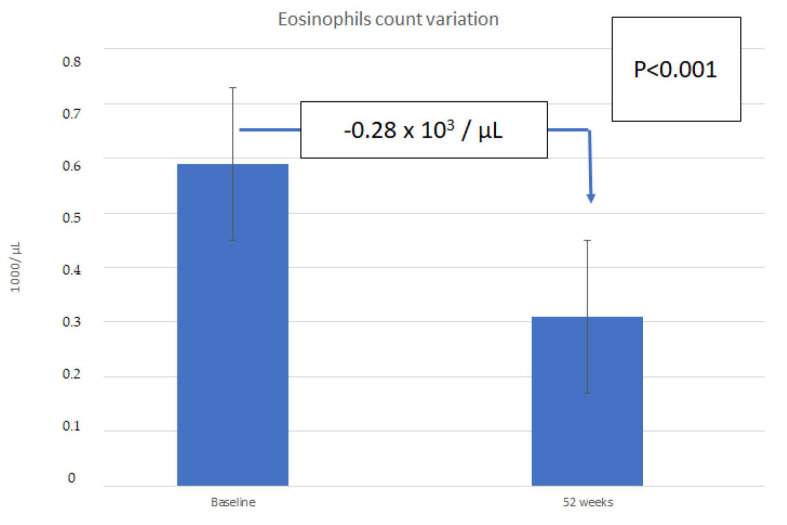
Change in circulating eosinophil count in children and adolescents with asthma (n = 12) after 52 weeks of treatment with Omalizumab.

**Table 1 children-08-01141-t001:** Baseline demographic and clinical data of children and adolescents with asthma before treatment with Omalizumab (n = 12).

Characteristics	Patients n = 12
Age (years), mean (SD)	12.4 (4.1)
Sex, n (%)	
Male	9 (75)
Age at diagnosis (years), mean (SD)	6.5 (3.77)
Personal history of atopy, n (%)	
Atopic dermatitis	9 (75)
Allergic rhinitis	10 (83.3)
Positive family history of atopy/asthma, n (%)	8 (66.7)
Total serum IgE (IU/mL), median (range)	1102.6 (371.7)
Eosinophil count, median (range) (×10^3^/mm^3^)	0.589 (0.1–0.99)
Allergic sensitization n (%)	
Polysensitization	11 (91.7%)
Monosensitization	1 (8.3%)
FEV1 (% of predicted), mean (SD)	86.74 (16.01)
MEF50 (% of predicted), mean (SD)	76.30 (27.22)
Number of asthma exacerbations in the previous year, before starting Omalizumab, mean (SD)	4.1 (2.8)
ICS dose at baseline (μg/day, fluticasone propionate equivalent)	
Mean (SD)	469.7 (199.84)
Median (range)	500 (250–1000)
Asthma long-term control medications at baseline, n (%)	
Leukotriene receptor antagonist	11 (91.7)
Long-acting β2-agonist	8 (66.7)
Oral corticosteroid	0 (none.)
Level of asthma control before treatment with Omalizumab, n (%)	
Controlled	0 (0)
Partial controlled	5 (41.7)
Uncontrolled	7 (58.3)
Treatment with Omalizumab, years, median (SD)	3 (2.094)

FEV1: forced expiratory volume in 1 s, MEF50: forced expiratory flow rate at 50% of forced vital capacity, ICS: inhaled corticosteroids.

**Table 2 children-08-01141-t002:** The effects on children and adolescents with severe asthma of one year of treatment with Omalizumab.

Parameter	Baseline	1 Year	*p*
Eosinophil count, median (×10^3^/mm^3^)	0.589	0.309	<0.0001
FEV_1_ (% of predicted), mean	86.74	105.03	<0.0001
MEF_50%_ (% of predicted), mean	76.30	102.13	<0.0001
Patients with ≥2 asthma exacerbations, n (%)	12 (100)	2 (16.7)	<0.0001
Number of asthma exacerbations after the first year of treatment with Omalizumab, mean (SD)	4.1 ± 2.8	1.15 ± 0.78	<0.0001
Level of asthma control before treatment with Omalizumab, n (%)			
Well-controlled	0 (0)	9 (75)	
Partial controlled	5 (41.7)	3 (25)	
Uncontrolled	7 (58.3)	0	
Decrease in maintenance doses of treatment, n (%)	-	9 (75)	

FEV1: forced expiratory volume in 1 s, MEF50: forced expiratory flow rate at 50% of forced vital capacity, ICS: inhaled corticosteroids.

## Data Availability

The data presented in this study are available upon reasonable request from the corresponding author.

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
