# Peer review of "Clinical Experience with Anti-IgE Monoclonal Antibody (Omalizumab) in Pediatric Severe Allergic Asthma—A Romanian Perspective"

_children, 2021, doi:10.3390/children8121141_

Round 1

Reviewer 1 Report

Berghea et al. present a retrospective observational real-life study of the efficacy and safety of Omalizumab in Romanian children with severe allergic asthma. The main strength of the study is the assessment of Omalizumab effectiveness (the real-life efficacy) in the population of Romanian children with severe atopic asthma. The main weakness of the study is the small number of included patients. It will be an advantage if the authors clarify the whole number of children with asthma in their database for the observed time period (2013-2019) and point the percentage of severe atopic asthma.

The "Materials and methods" section needs to be improved and specified- it is not clarified that the study is retrospective.

The methods should be described in more detail and clarity. How is asthma control assessed and evaluated – ACT (Asthma control test) or GINA criteria (ACT and/or C-ACT, row 180-181; GINA criteria, row 172-174), spirometry (reference equation, quality control, are all children able to perform good quality spirometry), does "eosinophil number" means peripheral blood eosinophils count - automated or manual?

The inclusion criteria should be clearly defined -row 122-126 (?). Is FEV1<80% inclusion criteria in the study? Please clarify.

In the "results" section - how many children are on control treatment with LTRA - 97,4% (text) or 91.7% (table 1), on LABA ? - row 167, on OCS - table I - 100% or 0%. The index used in figure 1 is not appropriate - the elevation of the absolute value of the FEV1 could be the result of the natural lung growth. FEV1 per cent predicted is a better and more objective choice for outcome index. 

Author Response

Berghea et al. present a retrospective observational real-life study of the efficacy and safety of Omalizumab in Romanian children with severe allergic asthma. The main strength of the study is the assessment of Omalizumab effectiveness (the real-life efficacy) in the population of Romanian children with severe atopic asthma. The main weakness of the study is the small number of included patients. It will be an advantage if the authors clarify the whole number of children with asthma in their database for the observed time period (2013-2019) and point the percentage of severe atopic asthma.

Authors’ comments:

Thank you for this suggestion. The authors added in the new manuscript the below sentence:

“The prevalence of severe allergic asthma in patients registered in the database of the ”M.S. Curie”, Children’s Emergency Clinical Hospital is 1.79%.” (lines: 108-110)

The "Materials and methods" section needs to be improved and specified- it is not clarified that the study is retrospective.

 Authors’ comment:

Thank you for this observation. The authors has corrected this error in the new version of the manuscript (line 18 and 104)

The methods should be described in more detail and clarity. How is asthma control assessed and evaluated – ACT (Asthma control test) or GINA criteria (ACT and/or C-ACT, row 180-181; GINA criteria, row 172-174), spirometry (reference equation, quality control, are all children able to perform good quality spirometry), does "eosinophil number" means peripheral blood eosinophils count - automated or manual? –

Authors’ comment:

Thank you for these comments. The authors changed the text in the "Materials and methods" section as you suggested. (please see the lines 103-162).

The inclusion criteria should be clearly defined -row 122-126 (?). Is FEV1<80% inclusion criteria in the study? Please clarify. –

Authors’ comments:

Thank you for your comment, we have clarified the inclusion criteria, as you suggested.  FEV1<80% was an inclusion criteria for patients aged between 12-18 years at the intiation of treatment with Omalizumab.

In the "results" section - how many children are on control treatment with LTRA - 97,4% (text) or 91.7% (table 1), on LABA ? - row 167, on OCS - table I - 100% or 0%. The index used in figure 1 is not appropriate - the elevation of the absolute value of the FEV1 could be the result of the natural lung growth. FEV1 per cent predicted is a better and more objective choice for outcome index. 

Authors’ comment:

Thank you for these observations. The authors corrected the unintentional errors in tables 1 and figure 1.

Reviewer 2 Report

The paper "Clinical experience with anti-IgE monoclonal antibody (Omalizumab) in paediatric severe allergic asthma – a Romanian perspective", by Berghea et al. provides a good analysis of the first national cohort of children with severe allergic asthma treated with Omalizumab. It is important to see how much improvement can be achieved in the real life cohorts of children treated with the new therapeutic options. But some issues can be improved. 

  1. Introduction. There are some redundant information that can be mention only  in the Material and Methods section. The definition of the severe asthma is well presented at page 3, line 113. The data about the approval of Omalizumab in Romania is mentioned at page 2 line 95 and in page 3 line 117. Because introduction is a bit long, I think you can kip it in the chapter 2.
  2. Page 2, line 64, line 73. No need for bold letters.
  3. Page 3, line 146. You need to add information about spirometry in children: ATS/ERS criteria?, type of spirometer, daily calibration? the best 3 out of possible 8 tests?, certified technician?, the same technician for the group? It is important to describe this sensitive issue in children.
  4. Page 3, line 130. Can you mention the definition of the exacerbation according to GINA? You can add a comment about the definition of the exacerbation. It is not clear if the patients had only severe exacerbations. 
  5. Page 4. line 152. I think you can mention here the ethical approval and inform content statement. Or can be kept as it is in page 9.
  6. Page 4, line 160. 75% boys. The cohort is small, but is there a male prevalence in the severe paediatric asthma? 
  7. Table 1. Very high level of IgE. Is there a particularity of the cohort?
  8. Table 2. Differences are highly significative at T1. Can you add a p?   
  9. Page 7, discussion. No need for bold parts of the text. 
  10. Page 8. No need for bold words. 

Author Response

The paper "Clinical experience with anti-IgE monoclonal antibody (Omalizumab) in paediatric severe allergic asthma – a Romanian perspective", by Berghea et al. provides a good analysis of the first national cohort of children with severe allergic asthma treated with Omalizumab. It is important to see how much improvement can be achieved in the real life cohorts of children treated with the new therapeutic options. But some issues can be improved. 

  1. Introduction. There are some redundant information that can be mention only  in the Material and Methods section. The definition of the severe asthma is well presented at page 3, line 113. The data about the approval of Omalizumab in Romania is mentioned at page 2 line 95 and in page 3 line 117. Because introduction is a bit long, I think you can kip it in the chapter 2.

Authors’ comments:

Thank you for these observations. The authors have modified the “Introduction” in the new reviewed manuscript. (please see the lines: 35-102

  1. Page 2, line 64, line 73. No need for bold letters.

Authors’ comments:

Thank you. We corrected it in the new version of the manuscript.

  1. Page 3, line 146. You need to add information about spirometry in children: ATS/ERS criteria?, type of spirometer, daily calibration? the best 3 out of possible 8 tests?, certified technician?, the same technician for the group? It is important to describe this sensitive issue in children. –

Authors’ comments:

Thank you for your suggestion. In the new version of the manuscript, we have added all these information (line 140 – 150)

  1. Page 3, line 130. Can you mention the definition of the exacerbation according to GINA? You can add a comment about the definition of the exacerbation. It is not clear if the patients had only severe exacerbations. 

Authors’ comments:

Thank you for your comment. In the new version of the manuscript, we have added the GINA definition (line 116 – 121)

  1. Page 4. line 152. I think you can mention here the ethical approval and inform content statement. Or can be kept as it is in page 9.

Authors’ comments:

Thank you for your comment. In order not to duplicate the information, authors decided to keep as it is.

  1. Page 4, line 160. 75% boys. The cohort is small, but is there a male prevalence in the severe paediatric asthma? 

Authors’ comments:

Thank you for your comment. The higher prevalence of boys in our cohort and the high prevalence of polysensitization are in line with the previous published data. It is described a tendency of higher prevalence of asthma in boys than in girls in childhood, a predominance of early-onset atopic asthma in male and a dominant proportion of male patients in paediatric patients with severe allergic asthma undergoing on Omalizumab treatment is also reported in other published studies (ref 20, 37, 40).

  1. Table 1. Very high level of IgE. Is there a particularity of the cohort?

Authors’ comments:

Thank you for your comment. We are sorry to cause confusion. We did not assume that the high level of total IgE is a particularity of our cohort  and we did not underline this result, we assummed that is a characteristic usual associated with paediatric severe allergic asthma as in other published studies.

  1. Table 2. Differences are highly significative at T1. Can you add a p? 

Authors’ comments:

Thank you for this observation. We have included p value in the table 2.

  1. Page 7, discussion. No need for bold parts of the text. 

Authors’ comments:

Thank you. We corrected it in the new version of the manuscript.

  1. Page 8. No need for bold words. 

Authors’ comments:

Thank you. We corrected it in the new version of the manuscript

General remark.

The authors asked the revision of the entire manuscript by an English native speaker to correct the language errors

We thank you for the useful suggestions and constructive report aiming at improving the scientific quality of our article. We agree with all the points raised and in the new revised version of our paper we have implemented modifications addressing these points.

We hope that the new version of our manuscript will be found above the acceptance standards of your journal

Round 2

Reviewer 1 Report

The authors took into consideration the comments and recommendations. The quality of the paper was obviously improved.

There are still some minor technical issues or mistakes. (e.g. line 215, 378-381, 384, 144-146…)

In lines 309-310 in the Results section, the authors report their data for mild reactions observed in the study group. In the same sentence, they cite already published results in the literature. In lines 423-425, there is a similar issue. On the other hand, in line 430, the citation is missing.

Author Response

Reviewer1

The authors would like to thank again the reviewer for his/her time and professional analysis of the manuscript. We are very grateful for all  comments, we have made the new necessary changes in the text.

The authors took into consideration the comments and recommendations. The quality of the paper was obviously improved.

There are still some minor technical issues or mistakes. (e.g. line 215, 378-381, 384, 144-146…)

Thank you for this observation. The authors has corrected the technical errors in the new version of the manuscript. In order to eliminate the superposed changes in the text, we sent the corrected manuscript and we have verified not to have doubled words or spelling mistakes (line 215  is line 171 in the final version, 378-381, 384, 144-146... is the Discussion section - corrected)

In lines 309-310 in the Results section, the authors report their data for mild reactions observed in the study group. In the same sentence, they cite already published results in the literature. In lines 423-425, there is a similar issue. On the other hand, in line 430, the citation is missing.

Thank you for the carreful analysis. We have corrected – the numbers in paranthesis are the number of patients with recorded adverse reactions, not citation (line 309 -310 , in the corrected manuscript 239-240).

The correspondent of lines 423-425 is now 338-340 – no need of citation at this lines, there are our results. 

Line 430 – is line 343 in the corrected manuscript – the citation is added.

We thank you for the useful suggestions and constructive report aiming at improving the scientific quality of our article.

We hope that the new version of our manuscript will be found above the acceptance standards of your journal.
